# Endoscopic Vacuum Therapy (EVT) versus Self-Expandable Metal Stent (SEMS) for Anastomotic Leaks after Upper Gastrointestinal Surgery: Systematic Review and Meta-Analysis

**DOI:** 10.3390/life13020287

**Published:** 2023-01-19

**Authors:** Francesco Vito Mandarino, Alberto Barchi, Ferdinando D’Amico, Lorella Fanti, Francesco Azzolini, Edi Viale, Dario Esposito, Riccardo Rosati, Gionata Fiorino, Willem Adrianus Bemelman, Ugo Elmore, Lavinia Barbieri, Francesco Puccetti, Sabrina Gloria Giulia Testoni, Silvio Danese

**Affiliations:** 1Division of Gastroenterology and Gastrointestinal Endoscopy, IRCCS San Raffaele Scientific Institute, Vita-Salute San Raffaele University, 20132 Milan, Italy; 2Department of Gastrointestinal Surgery, IRCCS San Raffaele Scientific Institute, Vita-Salute San Raffaele University, 20132 Milan, Italy; 3Department of Gastroenterology and Digestive Endoscopy, San Camillo-Forlanini Hospital, 00152 Rome, Italy; 4Department of Surgery, Cancer Center Amsterdam, 1081 HV Amsterdam, The Netherlands

**Keywords:** endoscopic vacuum therapy, esophagectomy, leakages, meta-analysis, stenting

## Abstract

Background: Endoscopic treatment of post-esophagectomy/gastrectomy anastomotic dehiscence includes Self-Expandable Metal Stents (SEMS), which have represented the “gold standard” for many years, and Endoscopic Vacuum Therapy (EVT), which was recently introduced, showing promising results. The aim of the study was to compare outcomes of SEMS and EVT in the treatment of post-esophagectomy/gastrectomy anastomotic leaks, focusing on oncologic surgery. Methods: A systematic search was performed on Pubmed and Embase, identifying studies comparing EVT versus SEMS for the treatment of leaks after upper gastro-intestinal surgery for malignant or benign pathologies. The primary outcome was the rate of successful leak closure. A meta-analysis was conducted, performing an a priori-defined subgroup analysis for the oncologic surgery group. Results: Eight retrospective studies with 357 patients were eligible. Overall, the EVT group showed a higher success rate (odd ratio [OR] 2.58, 95% CI 1.43–4.66), a lower number of devices (pooled mean difference [pmd] 4.90, 95% CI 3.08–6.71), shorter treatment duration (pmd −9.18, 95% CI −17.05–−1.32), lower short-term complication (OR 0.35, 95% CI 0.18–0.71) and mortality rates (OR 0.47, 95% CI 0.24–0.92) compared to stenting. In the oncologic surgery subgroup analysis, no differences in the success rate were found (OR 1.59, 95% CI 0.74–3.40, I^2^ = 0%). Conclusions: Overall, EVT has been revealed to be more effective and less burdened by complications compared to stenting. In the oncologic surgery subgroup analysis, efficacy rates were similar between the two groups. Further prospective data need to define a unique management algorithm for anastomotic leaks.

## 1. Introduction

Anastomotic dehiscences are defined as “full-thickness gastrointestinal defects involving esophagus, anastomosis, staple line, or conduit” by the Esophagectomy Complications Consensus Group (ECCG) [1], primarily due to anastomotic closure defects.

Dehiscence remains the most fearsome complication of upper gastrointestinal surgery [2,3] and is associated with high rates of postoperative morbidity and mortality [4,5,6]. Despite advances in surgical techniques, the incidence is still high (approximately 11.4% after esophagectomy [7] and 9% after gastrectomy [8]).

Management of the leak depends largely on the location. A post-esophagectomy cervical leak usually benefits from simple neck drainage and wound care [9]. The intrathoracic or intra-abdominal leak is more complex and treatment options vary from conservative treatment to surgical reintervention, depending on several factors, including the patient’s clinical condition, size of the leak, and timing since surgery [10]. With the aim of avoiding redo-surgery, endoscopic options have improved over time and to date are widely spread in common clinical practice.

Self-Expandable Metal Stents (SEMS) have represented the mainstay endoscopic technique for many years, promoting healing primarily due to mechanical bypass of the wall defect (Appendix A) [11]. Nevertheless, the use of SEMS is characterized by several complications, especially migration, which can occur in up to 23% of cases [12].

Since its advent in 2008 [13], EVT has proven to be a promising endoscopic technique for the management of surgical dehiscences. It consists of intra- or extra-luminally placed polyurethane foam connected via a tube to a vacuum device applying continuous negative pressure with the aim of cleaning the anastomotic site, promoting shrinkage of the para-anastomotic cavity (Appendix A). EVT is widely spreading as a treatment of post-esophagectomy/gastrectomy leaks, being increasingly preferred to stenting, as it is an effective and less-complicated approach; a recent prospective trial by Zhang et al. showed an 89% closure rate of post-esophagectomy intra-thoracic leakages, with a 5.4% rate of adverse events [14].

To date, few retrospective studies have compared the effectiveness of EVT and stenting in treating post-esophagectomy/gastrectomy leaks. The aim of this systematic review and meta-analysis is to summarize the evidence on the efficacy and adverse event occurrences of these two techniques, particularly focusing on oncologic surgery.

## 2. Materials and Methods

This systematic review was designed according to the guidelines of the Preferred Reporting Items for Systematic Reviews and Meta-analyses (PRISMA) checklist (Appendix A) [15].

The study protocol was registered and accepted in the International Prospective Register of Systematic Reviews (PROSPERO) [16], available at https://www.crd.york.ac.uk/prospero, accessed on 8th August 2022, (protocol code: CRD42022343885). No Institutional Review Board approval was needed.

### 2.1. Search Strategy and Study Selection

We performed a systematic literature search for published studies on Pubmed, Embase, and MEDLINE.

The primary search strategy was performed using tools implemented in the above-mentioned databases. For Pubmed, MeSH terms were included in the search string. The following keywords were used as MeSH: “Anastomotic Leak/prevention and control”; “Anastomotic Leak/surgery”; “Anastomotic Leak/therapy”; and “Upper Gastrointestinal Tract”.

Only manuscripts in English and studies published in peer-reviewed journals until June 2022 (cut-off date) were included.

The potential eligibility of articles was assessed by titles and abstracts. Then, the full texts of the manuscripts were analyzed, and the final decision for inclusion was made after a detailed review of the articles. Study selection was assessed by AB, and the evaluation of full texts of all relevant articles was performed by FVM and AB. Disagreements were resolved by consultation with a third investigator (LF). The detailed search strategy is shown in Appendix A.

### 2.2. Eligibility Criteria

We included prospective and retrospective clinical studies comparing EVT versus SEMS for the treatment of leaks after esophago-gastric resection with esophago-duodenal, esophago-jejunal, or esophago-gastric anastomosis for malignant or benign pathologies.

Exclusion criteria were aged <18 years, non-human studies, case reports, narrative and systematic reviews, studies on bariatric surgery, non-comparative studies, works providing a comparison between different endoscopic techniques from those analyzed, and studies not providing a full text.

### 2.3. Data Extraction

The following data were extracted: Authors, year of publication, country, study design, age, sex, neoadjuvant treatment, etiology of surgery, type of surgery, histology, resection rate, leak etiology, EVT type, placement technique, EVT pressure, time between EVT sessions, stent type and size, time between stent sessions, treatment duration (in days), number of sponges/stents used, success rate, short-term complications rate, dislocation of sponge/stent rate, Intensive Care Unit (ICU) admission (in days), hospitalization time, and in-hospital mortality rate.

### 2.4. Outcomes

The primary outcome was the rate of successful leak closure, regardless of the assessment method (endoscopic or radiological). Secondary outcomes were in-hospital mortality, ICU and hospitalization time (in days), treatment duration (in days), short-term complications rate, dislocation of sponge/stent rate, and number of endoscopic treatments (stent and sponge changes).

Short-term complications were considered those in the course of hospitalization. We counted short-term complications and sponge/stent dislocations per patient.

### 2.5. Risk of Bias

Two review authors (FVM and AB) assessed the risk of bias using the Newcastle–Ottawa Scale (NOS) for assessing the quality of non-randomized studies in a meta-analysis [17]. According to the score, all studies were evaluated from three perspectives: The selection of the study groups, the comparability of groups, and the ascertainment of the outcome. In case of disagreements, a resolution was reached by discussion with a third reviewer (LF).

### 2.6. Data Analysis

We used the statistical software package RevMan 5 (Review Manager version 5.3.5–Cochrane Collaboration, Oxford, UK) to execute statistical analysis, including the generation of forest plots and calculation of confidence intervals. Categorical data, means, and standard deviations were used for data analysis. For studies that did not report mean and standard deviation, the mathematical formula by Hozo et al. was used for data conversion [18].

A meta-analysis was conducted, not taking into account data homogeneity. For dichotomous variables, Odd Ratios (OR) were calculated using the Mantel–Haenszel test. The mean difference (MD) was used for continuous variables, determined by inverse variance. A confidence interval (CI) of 95% was set for both measures. The homogeneity of effect sizes among pooled studies was assessed with the I^2^ statistic and Chi-Square test (χ^2^). According to Higgins et al., I^2^ was interpreted as follows: Insignificant heterogeneity for I^2^ 0–25%, low heterogeneity for I^2^ 25–50%, moderate heterogeneity for I^2^ 50–75%, and high heterogeneity when I^2^ was greater than 75% [19]. Fixed or random effects were applied according to values of heterogeneity in each analysis (<50% or >50% values, respectively).

An a priori-defined subgroup analysis [20] was performed for the oncologic surgery group, in order to decrease clinical heterogeneity, and was implemented when at least 3 studies could be included.

## 3. Results

### 3.1. Study Selection

A total of 419 records were identified by the initial search strategy. After seven duplicate records were removed, a total of 412 records were screened, excluding a further 349 records due to non-compliance with the inclusion criteria. Sixty-three reports were assessed for eligibility and 55 were excluded as follows: Non-comparative studies (n 45), comparison studies with other treatments (n 7), unfulfilling data (n 2), and non-human studies (n 1). A total of eight retrospective studies were considered eligible for quantitative synthesis [21,22,23,24,25,26,27,28].

The whole selection and screening process (PRISMA flowchart) is synthesized in Figure 1.

### 3.2. Study Assessment

According to the NOS criteria, six studies [21,22,23,24,26,27] were assessed as fair quality with a moderate risk of bias, while the other two studies were evaluated as poor quality [25,28]. Details are shown in Appendix A.

### 3.3. Studies Description

Among the eight studies selected, seven were from Germany and one from Korea, with a total of 357 enrolled patients (152 patients in the EVT group and 205 patients in the stent group). Baseline characteristics are shown in Table 1, while endoscopic details are reported in Appendix A and outcomes in Table 2.

All studies only included patients with post-oncological esophagectomy and/or gastrectomy leakages, except for Brangewitz et al. [26] and Menningen et al. [27], which, in addition to post-oncological surgical leaks, also included leakages after surgery for benign diseases such as iatrogenic perforations and Boerhaave syndrome (n 14, 19.7% and n 2, 4.4%, respectively). Furthermore, all studies compared EVT and stent, except for Schniewind et al. [25] and Eichelmann et al. [28], which carried out a three-arm comparison between EVT, stent, and surgery in the management of post-esophagectomy leaks for malignant diseases.

All studies provided data on the success rate (primary outcome) in the original full-text manuscript, except for Schniewind [25] and Eichelmann [28]. Concerning secondary outcomes, the number of devices used and the duration of treatment were not reported by Schniewind et al. [25]; stent/sponge dislocation and short-term complication rates were not stated by Schniewind et al. [25], Menningen et al. [27], and Eichelmann et al. [28], while ICU admission time was not reported by Hwang et al. [23] and Mennigen et al. [27]. The in-hospital mortality rate was reported in all selected studies.

### 3.4. Publication Bias

Testing for publication bias was not necessary due to the limited number of studies included in the systematic review, according to Cochrane Collaboration recommendations.

### 3.5. Overall Outcomes

Overall, EVT showed a higher success rate compared to stenting (OR 2.58, 95% CI 1.43–4.66). All studies revealed the superior efficacy of EVT, except for Senne et al., which reported better results for esophageal stenting [24].

In terms of the duration of hospitalization, statistically significant differences were not found between the stent group and the EVT group (pooled mean difference of 5.46, 95% CI −3.87–14.79). The number of devices was lower for the stent group (pooled mean difference 4.90, 95% CI 3.08–6.71), while treatment duration was significantly shorter for patients treated with EVT (pooled mean difference −9.18, 95% CI −17.05–−1.32). Regarding adverse events, the EVT group was associated with a lower short-term complication rate (OR 0.35, 95% CI 0.18–0.71), while the dislocation rate was not different between the two groups (OR 0.63, 95% CI 0.28–1.41). Time in ICU did not differ between the EVT and stent groups (OR 1.32, 95% CI −2.99–5.63), while the in-hospital mortality rate was lower in the EVT group (OR 0.47, 95% CI 0.24–0.92).

Forest plots of overall outcomes are shown in Figure 2, Figure 3 and Figure 4.

### 3.6. Subgroup Analysis Outcomes

We performed an a-priori-defined subgroup analysis, analyzing only patients with post-oncologic esophagectomy and gastrectomy leaks.

In this subgroup analysis, in four studies [21,22,23,24], EVT and stent revealed no significant differences in the success rate (OR 1.59, 95% CI 0.74–3.40), showing insignificant heterogeneity (I^2^ = 0%) The duration of hospitalization was no different for the two techniques (pooled mean difference 7.98, 95% CI −3.39–19.36) (moderate heterogeneity, I^2^ = 52%)

The stent group showed a lower number of devices than the EVT group, with a pooled mean difference of 4.94 (95% CI 1.84–8.03) (high heterogeneity, I = 83%). We found that treatment duration was not statistically different between the two techniques (OR −9.07 95% CI −18.99, 0.85) (high heterogeneity, I^2^ = 77%), as well as the ICU admission time (pooled mean difference 1.32, 95% CI −2.99, 5.63) and in-hospital mortality rate (OR 0.57 95% CI 0.24, 1.37) (insignificant heterogeneity, I^2^ = 3% and 25%, respectively). The short-term complication rate was lower in the EVT group (OR 0.26, 95% CI 0.10, 0.68), showing insignificant heterogeneity (I^2^ = 0%), whereas the dislocation rate was not statistically different (OR 0.48, 95% CI 0.17, 1.37) (insignificant heterogeneity too, I^2^ = 0%). Forest plots of subgroup analysis outcomes are shown in Appendix A.

## 4. Discussion

SEMS and EVT represent the two most used techniques in the endoscopic treatment of anastomotic leakages after upper-gastrointestinal surgery; while the former still represents the “gold standard”, the latter has been providing promising results. Currently, comparison data between these two techniques are limited, and the choice between them varies across centers, depending on device availability and operator expertise.

In this scenario, this systematic review and meta-analysis had the aim of summarizing existing data on the efficacy and safety of these approaches, including eight retrospective studies (357 patients) comparing EVT and esophageal stenting. An a-priori-defined subgroup analysis of studies including only patients with anastomotic dehiscences after oncological upper gastrointestinal surgery (esophagectomies and gastrectomies) was performed, since Brangewitz et al. [26] and Menningen et al. [27] enrolled 14/71 and 4/75 patients with a non-malignant pathology, respectively.

The overall analysis revealed a higher efficacy (in terms of leakage closure) of EVT compared with stenting. In contrast, the subgroup analysis revealed no statistically significant differences in the success rate between EVT and stenting, with substantially lower heterogeneity (I^2^ = 0% versus I^2^ = 11%). These data are in line with most studies included in the analysis [21,22,23,24,25,28], as only Brangewitz et al. [26] and Mennigen et al. [27] revealed higher efficacy of EVT compared to the stent group. However, the non-uniform definition of success, generically described as the resolution of symptoms or radiological/endoscopic healing, could represent an important methodological issue in this analysis (Table 2).

Overall, the EVT group was associated with more devices used and lower mortality rates than the stent group. This is explained by the feature of the technique itself, consisting of scheduled short-time replacements (every 3–5 days) of the sponge, which leads to continuous monitoring of dehiscence evolution, preventing prolonged ineffective treatment. On the other hand, scheduled, close-in-time procedures could expose already frail patients to higher anesthesiologic risks, increasing treatment-related costs. Baltin et al. assessing the economic burden in the treatment of post-esophagectomy anastomotic leaks, showed that EVT is twice as expensive as stents (9.282 € versus 5.156 € per average case, respectively) [29]. However, the cost-effectiveness of the procedure is beyond the scope of our analysis. Nevertheless, subgroup analysis revealed no difference in mortality rates between the two techniques, even though heterogeneity was higher (I^2^ = 25% versus I^2^ = 0%, respectively).

EVT and esophageal stent placement represent safe procedures and major complications are quite rare [30]. Comprehensively, in our analysis, the short-term complication rate (*p* = 0.003) was higher for stenting versus EVT. This could be related to the technical difficulty of stent placement and the increased invasiveness of devices, leading to many burdensome adverse events such as bleeding, perforation, ulcers, ingrowth, and esophageal-tracheal fistulas. These results were confirmed in the sub-group analysis (*p* = 0.006). However, these complications were considered minor adverse events with a contained impact on patient survival. This is highlighted by in-hospital mortality rates in the sub-group analysis showing non-statistical differences. The higher mortality rate in the stent group in the overall analysis could be explained by the inclusion (by Brangewitz et al. [26]) of a greater number of severe conditions such as Boorehave syndromes, iatrogenic perforations, and descending aorta replacements.

Dislocations of devices were not included as short-term complications in our analysis and were assessed separately, comprehensively showing not statistically significant differences between the two groups. However, heterogeneous features of the devices used could represent a relevant confounding factor in the evaluation of this outcome, especially concerning the SEMS group: Four studies reported the use of both fully covered SEMS and partially covered SEMS [22,23,27,28], while Brangewitz et al. [26] used a plastic stent.

The results of our analysis must be interpreted with reserves, and some limitations must be acknowledged.

Firstly, the retrospective design of all the studies represents a relevant issue.

Moreover, data heterogeneity was a major limitation: Leaks did not have similar characteristics nor the same treatment management between the two groups (El Sourani et al. treated every leak with a related anastomotic cavity with EVT, whereas smaller defects with stenting [22]). In addition, Senne et al. managed all leakages before 2018 by SEMS placement, and after that year, by EVT [24].

Secondly, applied EVT techniques varied among different studies, affecting the standardization of the treatment. In five studies, the sponge was placed intracavitary [22,23,25,26,27], whereas Senne et al. [24] and Eichelmann et al. [28] also used the intraluminal technique if the size of the leak was reduced. Furthermore, negative pressure settings were different among studies. Schniewind et al. used a −80 mmHg pressure [25], while the other authors applied a pressure ranging from minus 100 to 125 mmHg [31]. However, the interval between sponge exchanges did not differ (3–5 days). Even sponge features were different: In five studies, the device was manufactured using a polyurethane sponge fixed to a naso-gastric tube [21,23,26,27,28], while the Esosponge (Braun, Aescula AG, Tuttlingen, Germany) device was used in the remaining three studies [22,24,25]. Nevertheless, the efficacy of EVT tended to be similar among studies, and the lower rates reported by Berthl et al. [21] and Brangewitz et al. [26] (71% and 84%, respectively) are attributable to larger patient cohorts (n 34 and n 32, respectively), while other studies enrolled a maximum of 15 patients in the EVT group [22,23,24,25,27,28].

Thirdly, different leak sizes between the two treatment groups are reported. In Brangewitz et al., 81% of leaks treated with EVT were at least 9 mm in diameter with cavities accessible by the scope, whereas 41% of leakages of the stent group were smaller than 9 mm [26]. Berth et al. reported more circumferential leaks in the EVT group compared to the stent group (*p* = 0.001) [21]. In El Sourani et al.’s study, the median size of leaks in the EVT group was larger than those in the stent group (15 mm versus 6 mm, respectively) [22]. In four studies, dehiscence sizes were not reported [24,25,27,28]. Since it is well known that leak size influences treatment outcomes, data heterogeneity may represent an additional bias in the comparison between the two treatments.

In terms of the primary outcome, our results are in line with two previous meta-analyses, which compared EVT and stenting in treating esophago-enteric anastomotic leakages [30] and upper gastrointestinal transmural defects [32], showing that EVT was associated with a higher rate of leak closure, more endoscopic device changes, a lower mortality rate, and shorter treatment duration. A recent update by Scognamiglio et al. confirmed his previous results [33]. Our meta-analysis provides further insight into endoscopic treatment of esophago-gastric leakages through a sub-group analysis including only studies with oncologic post-surgical anastomotic leakages, showing no statistical significance in terms of success rates between the two endoscopic treatments (insignificant heterogeneity, I^2^ = 0%), differently from previous reviews. However, these results might be biased by the above-mentioned limitations.

In our meta-analysis, the paradoxical discrepancy of efficacy results between overall and subgroup analyses, as well as the overall higher efficacy rate of EVT despite treating larger defects, deserves further evaluation.

Recently, a prospective study by Jung DH et al., assessing EVT in the management of anastomotic leaks, revealed the intraluminal technique (for small defects) as a predictor of clinical failure of the treatment [34]. In addition, Jung C. compared endoscopic internal drainage (EID) using plastic double pig-tail stents with EVT, reporting, for small leakages (without associated cavity), a slightly higher success rate of EID [35]. EID has shown higher efficacy rates even towards stenting for anastomotic leaks after esophageal/gastric resection, as shown by Hallitt et al. [36].

Therefore, regarding the state of the art, the optimal management of upper GI (UGI) leaks and perforations remains controversial, primarily due to the lack of studies comparing endoscopic techniques with defects of similar characteristics.

EVT is an emerging technique, providing promising results; future directions could include its use at the time of surgery for prophylactic purposes [37].

The above-mentioned most recent data and the results of our analysis might suggest how endoscopic techniques could all represent useful tools in the management of anastomotic leakages. The choice between them has to be tailored to leak characteristics such as size or the presence of a related cavity.

Additional data from future or ongoing prospective cohort studies [38] with larger populations, together with randomized controlled trials [39], need to clarify this hypothesis in order to identify a standardized management algorithm.

## Figures and Tables

**Figure 1 life-13-00287-f001:**
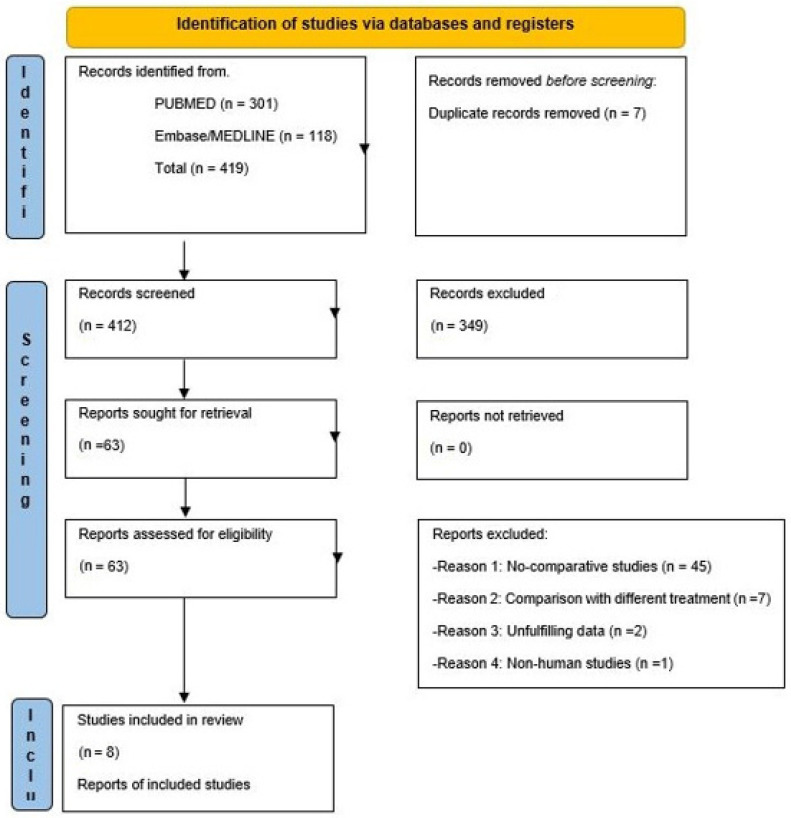
Flow diagram of study selection according to the Preferred Reporting Items for Systematic Reviews and Meta-analyses (PRISMA).

**Figure 2 life-13-00287-f002:**
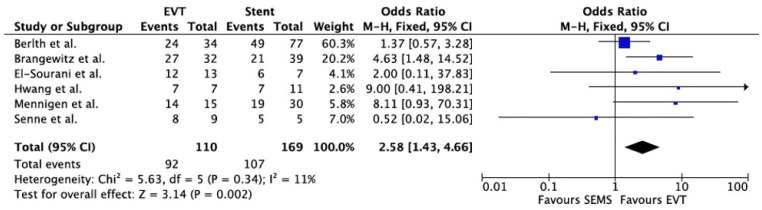
Forest plot for overall Treatment Success. Results are shown as Odd Ratio (OR) for categorical variables and as Mean Difference (MD) for continuous variables [21,22,23,24,26,27].

**Figure 3 life-13-00287-f003:**
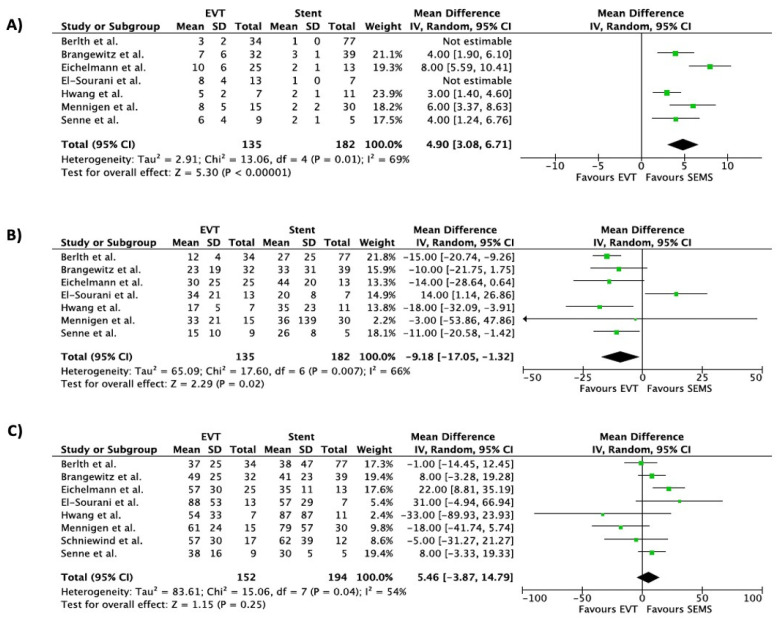
Forest plot for overall outcomes: (**A**) Number of devices, (**B**) treatment duration, (**C**) duration of hospitalization. Results are shown as Odd Ratio (OR) for categorical variables and as Mean Difference (MD) for continuous variables [21,22,23,24,25,26,27,28].

**Figure 4 life-13-00287-f004:**
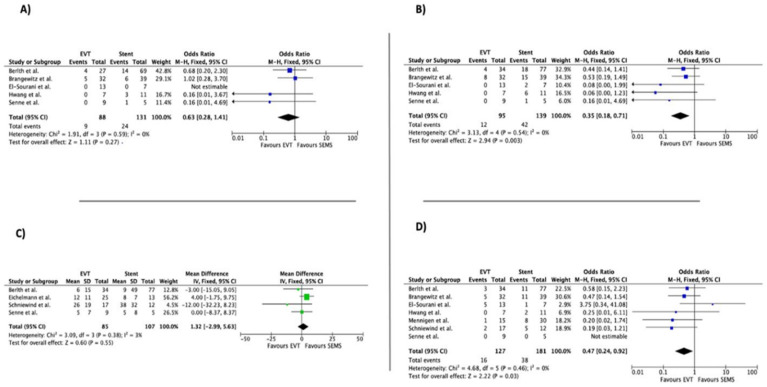
Forest plot for overall outcomes: (**A**) Dislocation, (**B**) short-term complication, (**C**) intensive Care Unit (ICU) time, (**D**) mortality. Results are shown as Odd Ratio (OR) for categorical variables and as Mean Difference (MD) for continuous variables [21,22,23,24,25,26,27,28].

**Table 1 life-13-00287-t001:** Baseline characteristics.

Author	Publication Year	Country	Study Design	Treatment	Patients (n)	Male, n (%)	Age, Median (Range)	Oncologic Resection n (%)	Neoadjuvant Therapy n (%)	Esophagectomies n (%)	Histology			Resection Rate	
											ADK	SCC	Other	R0	R1
Berlth et al. [21]	2018	Germany	Retrospective		111										
				EVT	34	29 (85)	65 (43–84)	34 (100)	26 (76.4)	25 (73.5)	28 (82.4)	5 (14.7)	1 (2.9)	N/A	N/A
				SEMS	77	63 (82)	64 (43–88)	77 (100)	50 (64.9)	68 (77.9)	46 (59.7)	29 (37.7)	2 (2.1)	N/A	N/A
Brangewitz et al. [26]	2013	Germany	Retrospective		71										
				EVT	32	28 (88)	65 (45–84)	28 (87.5)	18 (56.2)	14 (43.7)	N/A	N/A	N/A	N/A	N/A
				SEMS	39	30 (77)	62 (32–78)	29 (74.3)	6 (15.3)	27 (69.2)	N/A	N/A	N/A	N/A	N/A
El Sourani et al. [22]	2021	Germany	Retrospective		20										
				EVT	13	N/A	N/A	13 (100)	N/A	13 (100)	N/A	N/A	N/A	N/A	N/A
				SEMS	7	N/A	N/A	7 (100)	N/A	7 (100)	N/A	N/A	N/A	N/A	N/A
Hwang et al. [23]	2016	Korea	Retrospective		18										
				EVT	7	5 (71)	71.1 (63–78)	7 (100)	N/A	5 (71.4)	N/A	N/A	N/A	N/A	N/A
				SEMS	11	9 (82)	67.3 (55–81)	11 (100)	N/A	4 (36.3)	N/A	N/A	N/A	N/A	N/A
Menningen et al. [27]	2015	Germany	Retrospective		45										
				EVT	15	14 (93)	65.5 (40–92)	15 (100)	11 (73.3)	15 (100)	8 (53.3)	5 (33.3)	2 (13.3)	N/A	N/A
				SEMS	30	21 (70)	56 (42–76)	28 (93)	13 (43.3)	30 (100)	21 (70.0)	7 (23.3)	5 (33.3)	N/A	N/A
Schniewind et al. [25]	2013	Germany	Retrospective		35 *										
				EVT	17	N/A	N/A	17 (100)	N/A	17 (100)	N/A	N/A	N/A	N/A	N/A
				SEMS	12	N/A	N/A	12 (100)	N/A	12 (100)	N/A	N/A	N/A	N/A	N/A
Senne et al. [24]	2022	Germany	Retrospective		14										
				EVT	9	5 (56)	60 (36–79)	9 (100)	7 (77.7)	6 (66.6)	N/A	N/A	N/A	6 (66.6)	3 (33.3)
				SEMS	5	4 (80)	61 (21–79)	5 (100)	3 (60.0)	4 (80.0)	N/A	N/A	N/A	5 (100)	0 (0)
Eichelmann et al. [28]	2021	Germany	Retrospective		42 *										
				EVT	25	22 (88)	60 (42–78)	25 (100)	20 (80.0)	25 (100)	20 (80)	4 (16)	1 (4)	21 (84)	3 (12)
				SEMS	13	10 (77)	65 (37–88)	13 (100)	8 (61.5)	13 (100)	10 (77)	2 (15)	1 (8)	11 (85)	2 (15)

ADK, adenocarcinoma; SCC, squamocellular carcinoma; EVT, Endoscopic Vacuum Therapy; SEMS, Self-Expandable Metal Stents; N/A, Not Available data. * Total number of patients including surgical treatment.

**Table 2 life-13-00287-t002:** Outcomes.

Author	Treatment	Patients (n)	Definition of Success	Success, n (%)	Number of Device, Median (Range)	Treatment Duration (Days), Median (Range)	Duration of Hospitalization (Days), Median (Range)	Dislocation, n (%)	Short-Term Complications, n (%)	Time in ICU (Days), Median (Range)	In-Hospital Mortality, n (%)
Berlth et al. [21]		111	Endoscopical closure of leak, without signs of persistent dehiscence								
	EVT	34		24 (71)	3 (1–9)	12 (3–58)	37 (19–118)	4 (15)	0 (0)	6 (0–60)	3 (9)
	SEMS	77		49 (64)	1 (1–3)	27 (1–152)	38 (13–296)	14 (20)	4 (20)	9 (0–295)	11 (14)
Brangewitz et al. [26]		71	Radiological and endoscopic closure of leak, without clinical signs of persistent leakage, no leaks recurrence at follow-up								
	EVT	32		27 (84)	7 (5–28)	23 (9–86)	48.5 (21–122)	5 (16)	3 (9)	N/A	5 (16)
	SEMS	39		21 (54)	3 (2–6)	33 (9–132)	41 (2–93)	6 (15)	9 (23)	N/A	11 (28)
El Sourani et al. [22]		20	Complete closure of leak (assessment method not defined)								
	EVT	13		12 (92)	5 (4–18)	24.5 (8–80)	74 (9–193)	0 (0)	0	38 (9–193)	5 (38)
	SEMS	7		6 (86)	1 (1–2)	22 (3–31)	41 (22–123)	0 (0)	1	20 (16–57)	1 (14)
Hwang et al. [23]		18	Complete healing of leak, confirmed by EGD and X-ray								
	EVT	7		7 (100)	4.3 (2–10)	19.5 (5–21)	37.1 (13–128)	0 (0)	0	N/A	0 (0)
	SEMS	11		7 (86)	1.6 (1–4)	27 (3–84)	87.3 (17–366)	3 (27)	3 (27)	N/A	0 (0)
Menningen et al. [27]		45	Healing of anastomosis at endoscopic and X-ray evaluation								
	EVT	15		14/15 (93)	6.5 (1–18)	26.5 (3–75)	58 (23–106)	N/A	N/A	N/A	1 (7)
	SEMS	30		19/30 (63)	1 (1–6)	36 (1–560)	53 (13–195)	N/A	N/A	N/A	8 (27)
Schniewind et al. [25]		35 *	N/A								
	EVT	17		N/A	N/A	N/A	57 ± 30 (mean ± SD)	N/A	N/A	26 ± 19 (mean ± SD)	2 (12)
	SEMS	12		N/A	N/A	N/A	62 ± 39 (mean ± SD)	N/A	N/A	38 ± 32 (mean ± SD)	5 (42)
Senne et al. [24]		14	Endoscopic healing of dehiscence								
	EVT	9		8 (89)	6 ± 3.5 (mean ± SD)	14.8 ± 9.7 (mean ± SD)	38 ± 16 (mean ± SD)	0	0 (0)	4.8 ± 6.8 (mean ± SD)	0 (0)
	SEMS	5		5 (100)	2.4 ± 0.5 (mean ± SD)	26 ± 7.6 (mean ± SD)	30 ± 5 (mean ± SD)	1 (20)	0 (0)	5 ± 7.6 (mean ± SD)	0 (0)
Eichelmann et al. [28]		42	Defected cavity lined with surface epithelium at endoscopy and no leakage at X-ray								
	EVT	25		N/A	7.4 (1–25)	23 (3–101)	47 (14–119)	N/A	N/A	4 (1–37)	N/A
	SEMS	13		N/A	1.5 (1–3)	44 (11–68)	34 (17–56)	N/A	N/A	2 (1–26)	N/A

EVT, Endoscopic Vacuum Therapy; SEMS, Self-Expandable Metal Stents; ICU, Intensive Care Unit; N/A, Not Available data; EGD, esophagogastroduodenoscopy. * Total number of patients including surgical treatment.

## Data Availability

Data available on request due to restrictions eg privacy or ethical.

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
