# Peer review of "Endoscopic Vacuum Therapy (EVT) versus Self-Expandable Metal Stent (SEMS) for Anastomotic Leaks after Upper Gastrointestinal Surgery: Systematic Review and Meta-Analysis"

_life, 2023, doi:10.3390/life13020287_

Round 1

Reviewer 1 Report

This Italian group present a systematic review and metanalysis of the results for stenting versus vacum therapy for post-operative anastomotic leakage.

The manuscript is well-written; well-illustrated and well-discussed. I really do not have further comments on the text. 

I would like only to critize the dual first-authorship. This is only a subterfuge to increase the CV of 2 persons. Even twin brothers have differences and different legal rights based on primogeniture. Mandarino was include before Barchi for a reason (that is not obviously alphabetic).  

Author Response

We warmly thank the Reviewer for further evaluating our manuscript. 

As suggested, we proceeded to remove the co-authorship even if the two co-authors have equally contributed to the work

Reviewer 2 Report

Thanks for the opportunity to review the manuscript "Endoscopic vacuum therapy (EVT) versus self-expandable metal-stent for anastomotic leaks after upper GI-surgery: Systematic review and Meta-analysis" by Mandarino et al. The colleaques analysed 8 retrospective studies on the same topic with 357 patients included, and with the primary outcome of successflu leak closure. . Authors shows in the EVT group showed a higher success rate, a lower number of devices, a shorter treatment duration, a lower short-term complication, and mortality rates, compared to stenting. These are the relevant results of the metaanalysis and this is important to know for Upper GI-surgeons and advaced endoscopists. So, I would prefer the acceptance of the manuscript. Some minor mistakes have to be corrected:

Abstract: Start the Sentence with a big A in results.

Materials and Methods: Figure 1 has to be adopted, the headlines are not readable.

Discussion: Pleasew comment the following points: To perform a Metanalysis on 8 studies with retrospective character is not so meaningful. Moreover: to mix in this analysis the surgical procedures (esophagectomy and gastrectomy) decrease the result even more.

Author Response

We warmly thank the Reviewer for further evaluating our manuscript. 

  1. We have proceeded to correct the orthographic mistake in the sentence from the “Results” section of the abstract
  2. We adapted the headings of table 1 in the track version of the revised manuscript
  3. We agree with the reviewer suggestion. We are aware that so far available studies comparing EVT and SEMS for the treatment of anastomotic leaks are retrospective and present weaknesses in methodology. Nonetheless, we tried to summarize the available evidence with the current paper. We have performed a sub-group analysis analyzing only outcome of leaks after oncologic surgery in order to minimize biases. To best to our knowledge no previous studies focused on this concern. We agree also with the concern about the inclusion in the study of different surgical techniques (gastrectomies and esophagectomies). However, available studies comparing EVT and SEMS enrolled both types of surgical interventions, so the inclusion of all samples was the only feasible way to compare the two endoscopic techniques by met-analytic approach. Furthermore, studies including gastrectomies represent a minority of the whole sample.